# Systems genetics analysis of the LXS recombinant inbred mouse strains:Genetic and molecular insights into acute ethanol tolerance

**Richard A. Radcliffe**[1,2]*, **Robin Dowell**[3,4,5], **Aaron T. Odell**[3], **Phillip A. Richmond**[3], **Beth Bennett**[1], **Colin Larson**[1], **Katerina Kechris**[6], **Laura M. Saba**[1], **Pratyaydipta Rudra**[7], **Shi Wen**[6]

**1** Skaggs School of Pharmacy and Pharmaceutical Sciences, University of Colorado Anschutz Medical Campus, Aurora, CO, United States of America, **2** Institute for Behavioral Genetics, University of Colorado Boulder, Boulder CO, United States of America, **3** BioFrontiers Institute, University of Colorado Boulder, Boulder, CO, United States of America, **4** Department of Molecular, Cellular, and Developmental Biology, University of Colorado Boulder, Boulder, CO, United States of America, **5** Department of Computer Science, University of Colorado Boulder, Boulder, CO, United States of America, **6** Department of Biostatistics and Informatics, Colorado School of Public Health, University of Colorado Anschutz Medical Campus, Aurora, CO, United States of America, **7** Department of Statistics, Oklahoma State University, Stillwater, OK, United States of America

* richard.radcliffe@cuanschutz.edu

**Data Availability Statement:** The raw RNA-seq data and the processed data are available from the

## Abstract

We have been using the Inbred Long- and Short-Sleep mouse strains (ILS, ISS) and a recombinant inbred panel derived from them, the LXS, to investigate the genetic underpinnings of acute ethanol tolerance which is considered to be a risk factor for alcohol use disorders (AUDs). Here, we have used RNA-seq to examine the transcriptome of whole brain in 40 of the LXS strains 8 hours after a saline or ethanol "pretreatment" as in previous behavioral studies. Approximately 1/3 of the 14,184 expressed genes were significantly heritable and many were unique to the pretreatment. Several thousand *cis*- and *trans*-eQTLs were mapped; a portion of these also were unique to pretreatment. Ethanol pretreatment caused differential expression (DE) of 1,230 genes. Gene Ontology (GO) enrichment analysis suggested involvement in numerous biological processes including astrocyte differentiation, histone acetylation, mRNA splicing, and neuron projection development. Genetic correlation analysis identified hundreds of genes that were correlated to the behaviors. GO analysis indicated that these genes are involved in gene expression, chromosome organization, and protein transport, among others. The expression profiles of the DE genes and genes correlated to AFT in the ethanol pretreatment group (AFT-Et) were found to be similar to profiles of HDAC inhibitors. *Hdac1*, a *cis*-regulated gene that is located at the peak of a previously mapped QTL for AFT-Et, was correlated to 437 genes, most of which were also correlated to AFT-Et. GO analysis of these genes identified several enriched biological process terms including neuron-neuron synaptic transmission and potassium transport. In summary, the results suggest widespread genetic effects on gene expression, including effects that are pretreatment-specific. A number of candidate genes and biological functions were identified that could be mediating the behavioral responses. The most prominent of these was *Hdac1*

Gene Expression Omnibus (https://www.ncbi.nlm.
nih.gov/geo/), accession number GSE157215.

**Funding:** RAR: R01AA016957; National Institute of
Alcohol Abuse and Alcoholism; https://www.niaaa.
nih.gov/ LMS, KK: P30DA044223; National
Institute of Drug Abuse; https://www.drugabuse.
gov/ LMS: R24AA013162; National Institute of
Alcohol Abuse and Alcoholism; https://www.niaaa.
nih.gov/ KK: R01AA021131; National Institute of
Alcohol Abuse and Alcoholism; https://www.niaaa.
nih.gov/ The funders had no role in study design,
data collection and analysis, decision to publish, or
preparation of the manuscript.

**Competing interests:** The authors have declared
that no competing interests exist.

which may be regulating genes associated with glutamatergic signaling and potassium
conductance.

## Introduction

It has been well established that genetics is a contributing factor to the development of alcohol
use disorders (AUDs), yet knowledge of the underlying genetic variants is limited [1]. The
genetic analysis of human AUDs is complicated by a number of factors including low pene-
trance, population heterogeneity, poor control or lack of understanding of relevant non-
genetic factors, and the likelihood that dozens if not hundreds of small effect genes are
involved [2]. To some extent, these issues can be mitigated through the study of *endopheno-
types*; i.e., intermediate traits that are heritable, biologically plausible, and predictive of the
condition [3]. One well-studied endophenotype is acute ethanol sensitivity first noted by
Schuckit [4] who observed that individuals who had a family history of AUDs were reliably
less sensitive to an acute ethanol challenge compared to those without a family history, a phe-
nomenon Schuckit termed "low level of response" (LR; [5]). LR subsequently has been shown
to be heritable [6] and to be a reliable predictor of future drinking problems [5].

Acute functional tolerance (AFT) has been postulated to be an important component of
acute ethanol sensitivity, particularly in the context of the Schuckit LR hypothesis [7]. AFT was
first described by Mellanby [8] who observed that intoxicated dogs were more ataxic on the
rising limb of ethanol distribution than they were at the same blood ethanol concentration
(BEC) on the falling limb indicating the presence of AFT. Genetic effects on AFT have been
described in humans and a variety of model organisms [9–12]. AFT thus has been postulated
to be an important factor in the genetic relationship between acute sensitivity and AUD risk,
although this has not been firmly established [7, 13, 14].

We have been investigating the Inbred Long and Short Sleep selected mouse strains (ILS,
ISS) to understand the genetic and molecular underpinnings of acute sensitivity to ethanol
and AFT, and their relationship to drinking behavior. The ILS and ISS were selectively bred
for extreme differences in the duration of the loss of the righting response (LORR), colloquially
referred to as "sleeping time" (ST), which is an acute response to a hypnotic dose of ethanol
[15–17]. The ILS/ISS difference in ST is due primarily to a difference in neuronal sensitivity to
ethanol rather than to a metabolic difference which is confirmed by the approximately two-
fold difference between the ILS and ISS in BEC at the regain of the righting response [12]. ST
is one of the most common behavioral tests for ethanol sensitivity in rodents, yet AFT is typi-
cally not considered because it is technically difficult to obtain an accurate estimate of BEC at
the loss of function on the rapidly rising limb of ethanol distribution; an increase in BEC from
the loss to the regain of the righting response would indicate the development of AFT. To miti-
gate this shortcoming, Ponomarev and Crabbe [18] developed a variation of the original stan-
dard test designed by McClearn [17] with which it is possible to measure AFT more
accurately. Using this method, it was shown that a substantial portion of the enormous ILS/ISS
difference in ST was mediated by AFT [12].

We have extended our AFT studies using a recombinant inbred (RI) mouse strain panel
derived from the ILS and ISS, the LXS. The LXS panel was created from pairs of ILS/ISS-
derived F$_2$ offspring that were bred through brother-sister matings for more than 20 genera-
tions resulting in a panel of some 60 inbred strains, each of which contains a random assort-
ment of alleles from the ILS/ISS progenitors [19]. RI panels are often referred to as "reference"
populations with which phenotypic, genetic, and genomic results in distinct cohorts that have

been tested at different times or in different labs can be compared and co-analyzed [20]. We have been using a procedure with which mice are administered a "pretreatment" dose of ethanol or saline 24 hours before being tested for AFT. This procedure was originally employed to investigate rapid (one day) tolerance and we subsequently discovered that the pretreatment altered AFT in a genotype-dependent manner ([12]; for further discussion of the relationship between AFT and rapid tolerance, see [21]). A significant genetic correlation was found between AFT and drinking in the dark (DID), a mouse model of binge drinking, with higher drinking strains tending to have higher AFT, consistent with human studies [21]. Importantly, the correlation was significant only in the ethanol-pretreated cohort suggesting that one's baseline AFT may be less important as a risk for pathological drinking behavior than the way AFT changes in response to prior experience with ethanol. We have also mapped a significant quantitative trait locus (QTL) for AFT on distal mouse chromosome 4 at the same position that others have mapped drinking behavior in the mouse, including DID in the LXS strains [22–24]. As with the AFT/DID correlation, the QTL was specific for the ethanol-pretreated cohort. This locus may be a key in understanding the relationship between AFT and drinking behavior in the LXS RIs.

In a continuing effort to understand the genetic factors that contribute to AFT, here we report a "genetical genomics" experiment in which we have profiled the brain transcriptome using quantitative RNA sequencing (RNA-seq) in 40 of the LXS RI strains following saline or ethanol pretreatment. The 14,184 genes that were reliably expressed above background displayed a broad range of heritabilities and as much as a 70-fold difference in expression between the lowest and highest expressing strains. The ethanol pretreatment affected the expression of 1,230 genes in a strain-dependent manner changing the expression of some by as much as 3-fold. Thousands of *trans-* and *cis*-acting expression quantitative trait loci (eQTLs) were mapped with some of the *cis*-eQTLs accounting for nearly 100% of the variance in expression of their associated gene. Moreover, a substantial number of the eQTLs were specific to the pretreatment. Overall, the results begin to develop a mechanistic framework for the genetic basis of variation in AFT in the LXS RI strains.

## Materials and methods

### Animals

LXS RI breeders were obtained from The Jackson Laboratory (Bar Harbor, ME) and bred in-house in the University of Colorado Anschutz Medical Campus (UCAMC) vivarium, a pathogen-free facility. Offspring were weaned and sex-separated at 21 days of age. All experiments were conducted with males that were group-housed in standard housing containing from 2 to 5 mice per cage. Only male mice were used for reasons of economy and for breeding maintenance. To our knowledge, female ILS, ISS, or LXS have never been tested for ST AFT. Sex differences have been noted for ST in the LXS lines [25], but not for ST AFT among 21 inbred mouse strains [26]. The mice were between 56 and 106 days of age at the time of their use (mean = 80 ± 0.3). They were maintained in a constant temperature (22-23˚C), humidity (20-24%), and light (14L/10D) environment. The procedures described in this report have been established to ensure the absolute highest level of humane care and use of the animals, and have been reviewed and approved by the University of Colorado Anschutz Medical Campus Institutional Animal Care and Use Committee.

### RNA extraction and library preparation

All RNA-seq procedures have been published previously [27, 28]. Briefly, mice were administered normal saline (0.01 ml/g, ip) or ethanol in saline (5 g/kg, ip; 20% v/v) and sacrificed 8

hours later by $CO_2$ inhalation followed by decapitation. Here, we refer to the saline and ethanol administration as a "pretreatment" to be consistent with the behavioral studies [29]. The brain was removed and dissected into cerebellum and whole brain (minus the olfactory bulbs), and stored in RNALater at -20˚C until RNA extraction. The RNA-seq studies reported here used only the whole brain sample. Total RNA was extracted using RNeasy Mini kits (Qiagen, Valencia, CA), and quantity and quality were determined using a NanoDrop™ spectrophotometer (Thermo Fisher Scientific, Wilmington, DE) and Agilent 2100 BioAnalyzer™ (Agilent Technologies, Santa Clara, CA). Ratios of absorbance at 260nm and 280nm were shown to be excellent (>1.8) and RNA Integrity scores were also shown to be excellent (>8.0). Total RNA was stored at -80˚C until library preparation.

Total RNA was isolated from 9 mice per strain and an equal amount of RNA from 3 mice of the same strain and treatment condition was pooled for each library; thus, 3 libraries per strain were prepared. Pooling in this manner reduces within-strain variance producing an effective increase in statistical power without increasing the number of libraries [30]. Samples were enriched for poly-A RNA using the Dynabeads mRNA Purification kit (Invitrogen) as directed by the manufacturer. Paired-end (2x100, expected size of 300 bp), strand-specific, cluster-ready libraries were prepared from the poly-A enriched RNA using the ScriptSeq RNA-Seq Library Preparation Kit v2 (Illumina) following the manufacturer's instructions. Sequencing was performed by the University of Colorado Anschutz Medical Campus Genomics and Microarray Core on an Illumina HiSeq 2000 Sequencing System as per the manufacturer's instructions with 6 bar-coded libraries pooled per flow-cell lane. Up to 8 libraries were prepared at a time and within each library prep group, a saline and ethanol sample were paired for any given strain. With the exception of two occasions, there was never more than one saline/ethanol pair for any strain within each library prep group. A similar approach was used for sequencing and, with the exception of 5 cases, saline/ethanol pairs for any given strain were represented only once on an entire flow cell. This quasi-randomization procedure was implemented to minimize batch effects. In total, 250 libraries were prepared representing saline and ethanol treated mice from 42 LXS RI strains. A total of 14 libraries were removed due to poor quality or other technical issues. As a result, four strains (two in the saline group and two in the EtOH group) were left with an n of only one. These samples were also excluded leaving 40 strains in each of the pretreatment groups with n=2 and n=3 for 8 and 72 strain/pretreatment combinations, respectively. Mapping metrics can be found in S1 Table.

## Alignment, transcript assembly, and quantification

RNA-seq reads were mapped back to RI strain-specific genomes using TopHat2 (v2.06; [31]) using their respective transcriptome annotation files. TopHat2 was run in very sensitive mode, allowing for microexons but not novel junctions. Whole gene quantification was determined using HTSeq [32] which provides raw read counts over an annotated gene set. Only uniquely mapping reads were used for quantification.

For determination of genes that were expressed above background, a RPLG (Reads Per Length of Gene) value was calculated for each gene in each individual sample as follows: RPLG = (reads * 100) / gene length [33]. Given that the read length was 100 bp, a RPLG value of 1 indicates sequencing coverage of 1X on average for any given gene. The median RPLG was then calculated across samples within each pretreatment group for each gene; genes with a median RPLG greater than 1 in either the saline or ethanol pretreatment group were retained for further analysis. DESeq2 was used to normalize raw read counts to library size and transformed to achieve an approximately normal distribution using a variance stabilization transformation method (VST; [34]).

## Determination of ethanol-responsive genes

Initially, a two-way ANOVA (strain-by-pretreatment) was conducted to identify genes that were differentially expressed (DE) due to ethanol pretreatment. Even with a very conservative false discovery rate (FDR<0.001; [35]), nearly 10,000 of the genes were found to have a significant pretreatment effect. Since our goal was to identify high-confidence DE genes, we decided to take a simpler, more conservative approach. DE was tested in each individual strain using a one-way ANOVA with pretreatment as the factor (FDR<0.05). Complete results of the one-way and two-way analyses can be found in S3 Table.

## eQTL mapping and genetic correlation analysis

Genotype data for eQTL mapping were collected by Dr. Gary Churchill and colleagues at The Jackson Laboratory using the Affymetrix Mouse Diversity Genotyping Array (http://cgd.jax.org/mda/v1). Out of 314,865 SNPs on the array, 43,870 were of high quality and informative in the LXS. The final list of markers used for eQTL mapping consisted of 2,661 non-redundant SNPs.

Strain means for each gene within each pretreatment condition were entered into QTL Reaper, the batch-mode version of the mapping software used in the online tool GeneNetwork (www.GeneNetwork.org). QTL Reaper performs whole genome interval mapping using the Haley-Knott regression method. Significance thresholds were determined independently for each gene using the permutation method of Churchill and Doerge [36] with 1,000 permutations. The genome-wide significance threshold for eQTLs was considered to be p<0.05 and is expressed in the text, tables, and figures as the likelihood ratio statistic (LRS); permutated significance thresholds for p<0.001 and p<0.10 can be found in S3 Table.

eQTLs found within 5 Mb of the gene's location were considered to be *cis*-regulated; all others were considered to be *trans*-regulated. A conservative procedure was used to identify high confidence, pretreatment-unique eQTLs; these are referred to as "golden" eQTLs. *cis*-eQTLs as defined above were considered golden when there were no other peaks (p<0.10) in the other condition within 25 Mb of the gene's location. *trans*-eQTLs were considered golden when: 1) the gene and the peak were greater than 25 Mb away from each other when on the same chromosome and 2) there were no peaks in the other condition (p<0.10) within 25 Mb of the first peak regardless of whether the it was on the same or different chromosome.

Pearson product-moment correlations (*r*) on strain means between behavior and gene expression, or between expression and expression were conducted to determine the extent of shared genetic variance between two traits. Unless otherwise stated, a correlation was considered significant at FDR<0.10. Behavioral data are from Bennett *et al.* [22]. Here, we use the same abbreviations as in that study: ST-sal and ST-Et (duration of the loss of the righting response after saline or ethanol pretreatment, respectively); AFT-sal and AFT-Et (acute functional tolerance after saline or ethanol pretreatment, respectively).

## Heritability

Broad sense heritabilities ($H^2$) for the LXS transcripts were calculated using a linear mixed model applied to the data after using a Variance Stabilizing Transformation (VST) as previously described [28]. Significance of heritability was determined by testing the null hypothesis of no heritability using a likelihood ratio test [28]. Note that sample pooling reduced within-strain variance which likely had the effect of overestimating heritability.

## Enrichment analysis

Gene Ontology (GO) enrichment analysis was conducted using the online bioinformatics resource DAVID [37]. Genes were entered as gene names and the analysis was conducted using the default DAVID *Mus musculus* background. Terms from *Biological Process* (GOTERM_BP_ALL) and *Cellular Component* (GOTERM_CC_ALL) were considered significant at FDR<0.05. The online tool REVIGO was used to illustrate the DAVID output [38]. REVIGO uses a semantic similarity algorithm to cluster similar terms, shown as bubbles, in close proximity to one another revealing key themes in which input genes are putatively involved. To highlight the more biologically informative terms, the plots were constructed with smaller bubbles indicating broad, general terms and larger bubbles indicating more specific terms. The lists of DE genes and correlated genes were also examined for overlap with cell-specific gene expression signatures of neurons, astrocytes and oligodendrocytes using the online tool Gene Set Enrichment Analysis (GSEA; www.gsea-msigdb.org/gsea/index.jsp).

The online CLUE analysis environment (CMap Linked User Environment; https://clue.io) was used to investigate the possibility that the expression profiles generated from the correlation and DE analyses were similar to profiles resulting from other experimental treatments. The CLUE query tool searches over 1.3M L1000 gene expression profiles found in CMap (Connectivity Map) for compounds or genetic manipulations ("perturbagens") that produce an expression profile similar to a user-provided input profile [39, 40].

The input profiles were the DE genes or the genes correlated to each of the behaviors. CLUE only accepts human gene identifiers. Gene names were converted from mouse to human using the dbOrtho tool (https://biodbnet-abcc.ncifcrf.gov/) and entered as either up- or down-regulated; the up/down input for the DE gene list is obvious while genes whose expression was positively correlated to behavior were considered to be up-regulated and negatively correlated genes were considered to be down-regulated for the correlated gene inputs (note that the query tool examines the L1000 profiles in both directions). Each perturbagen was assigned a standardized value ranging from -100 to 100 called a "connectivity score" (τ) based on a summary of the similarity between the input profile and all of the perturbagen's profiles (each perturbagen is assayed using multiple doses and time points, and in up to 9 different cell lines). Perturbagen classes (PCLs) – groupings of perturbagens with similar mechanistic or biological activity that also show highly similar expression profiles – were also assigned a connectivity score.

## Expression data availability

The raw expression data and the VST-normalized DESeq2 output are available through the Gene Expression Omnibus (www.ncbi.nlm.nih.gov/geo/; accession number: GSE157215).

## Results

Using a moderately conservative filtering procedure (median RPLG>1 in either the saline group or the ethanol group), we found that 13,460 and 14,170 out of 38,087 Ensembl genes were expressed in the saline and ethanol cohorts, respectively. The union of these two gene sets was 14,184 genes; this gene list was used for all subsequent analyses (basic annotations can be found in S3 Table). There were 724 genes that were unique to the ethanol group and 14 genes were unique to the saline group. The difference may have been due to the overall greater number of raw reads in the ethanol group since low-expressing genes would be sensitive to the median RPLG filter. In fact, the mean (± SEM) RPLG for the 724 ethanol-unique genes was 0.91 (± 0.01) and 1.31 (± 0.02) in the saline and ethanol groups, respectively, indicating that

these genes were generally low-expressers. Similarly, the mean (SEM) RPLG for the 14 saline-unique genes was 1.28 (± 0.09) in the saline group and 1.08 (± 0.09) in the ethanol group.

Broad sense heritability for gene expression ranged from 0.00 to 0.99 in the saline pretreatment group and 2,587 of the heritabilities were found to be significant (FDR<0.05; S3 Table); the minimum significant heritability was 0.34. Heritabilities in the ethanol pretreatment group also ranged from 0.00 to 0.99 and the minimum significant heritability was 0.33; however, heritability was significant for approximately 85% more genes in the ethanol group (4,776) compared to the saline group (2,587; Fig 1). Additionally, heritability was generally higher in the ethanol group compared to the saline group; i.e., of the 2,309 genes that had significant heritability in both pretreatment groups, ethanol heritability was greater than saline heritability for 1,630 genes (71%; genes above the diagonal in Fig 1).

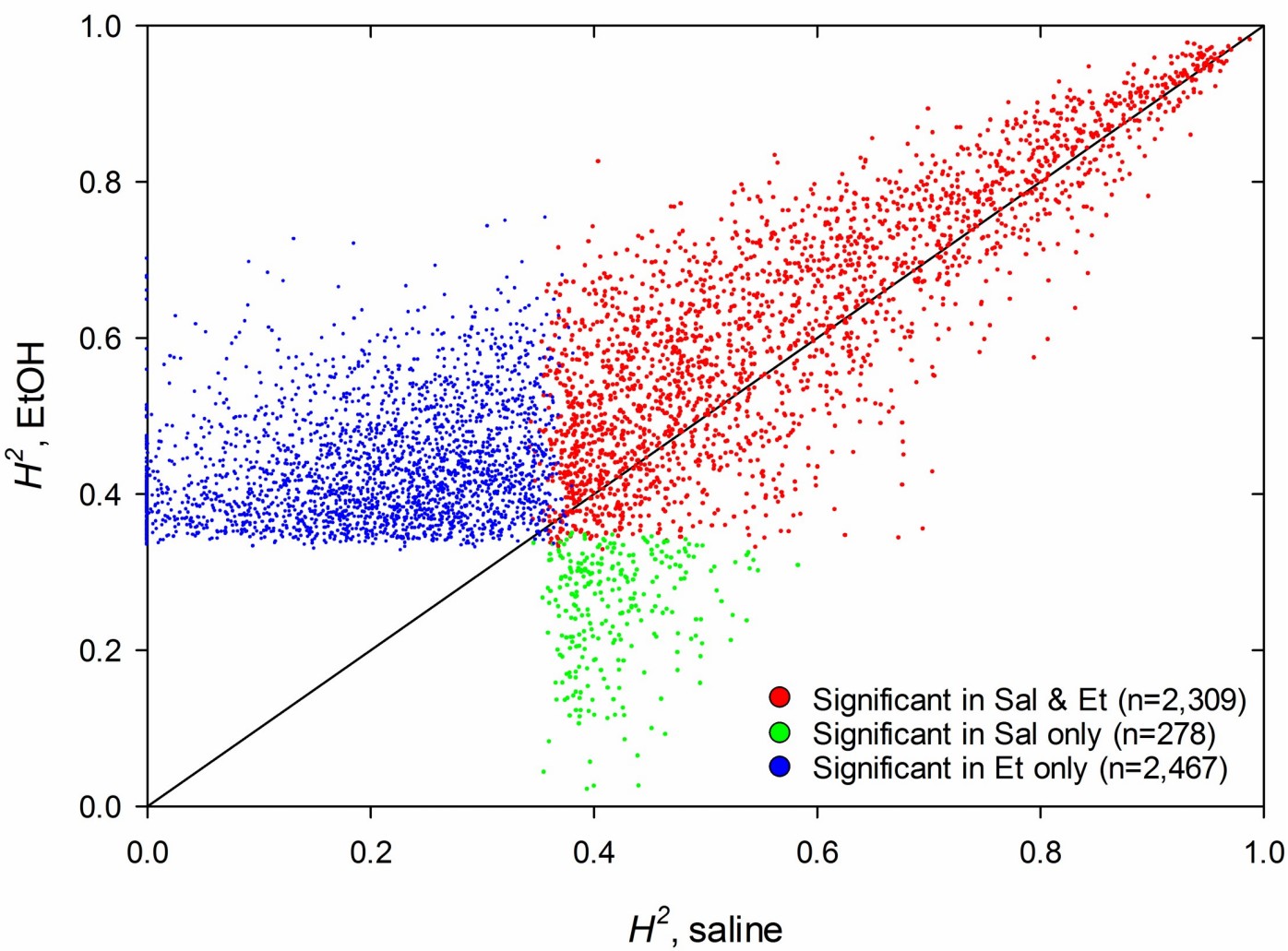

**Fig 1. Heritability ($H^2$) of expression for individual genes in the LXS RI panel.** Only genes that were significantly heritable in the saline and/or ethanol (Et) pretreatment are shown (FDR<0.05; n=5,054). Each dot represents a single gene. Red dots: significant in both pretreatment groups; green dots: significant in saline only; blue dots: significant in ethanol only.

A total of 1,230 genes were found to be DE due to the ethanol pretreatment in at least one strain and the number of altered genes in a single strain ranged from 0 to 551 (FDR<0.05; Fig 2A). The majority of genes were found in the top 5 strains (63%) with strain number 100 alone accounting for approximately 35% of the genes; *i.e.*, those genes that were expressed in only a single strain. Most of the genes were significant in only a single strain (1,007; 82%); the remainder (223; 18%) were significant in from 2 to 6 strains. The ethanol pretreatment caused 656 of the genes to be up-regulated (53%) and 572 genes to be down-regulated (46%); 2 genes were up- or down-regulated dependent on strain. Genes were up-regulated by 1.03- to 3.20-fold or down-regulated by 1.04- to 1.94-fold, and less than 6% of the genes were up- or down-regulated by 1.10-fold or less (Fig 2B).

A.

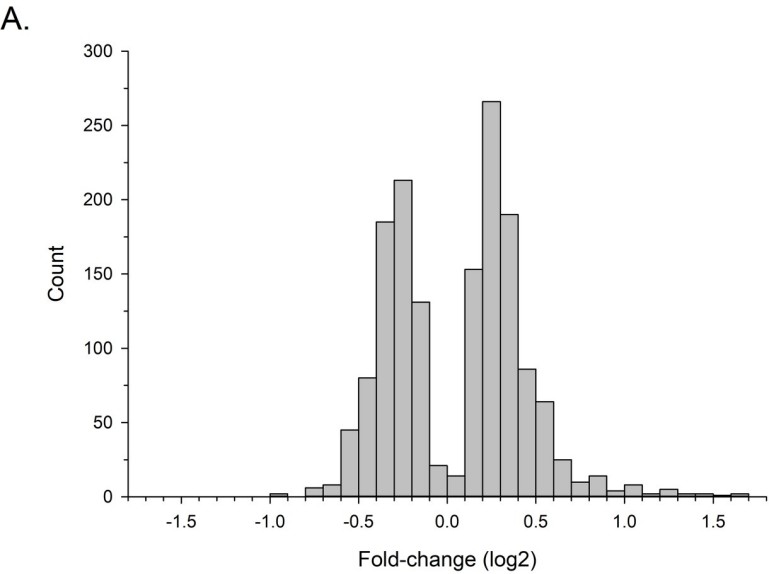

B.

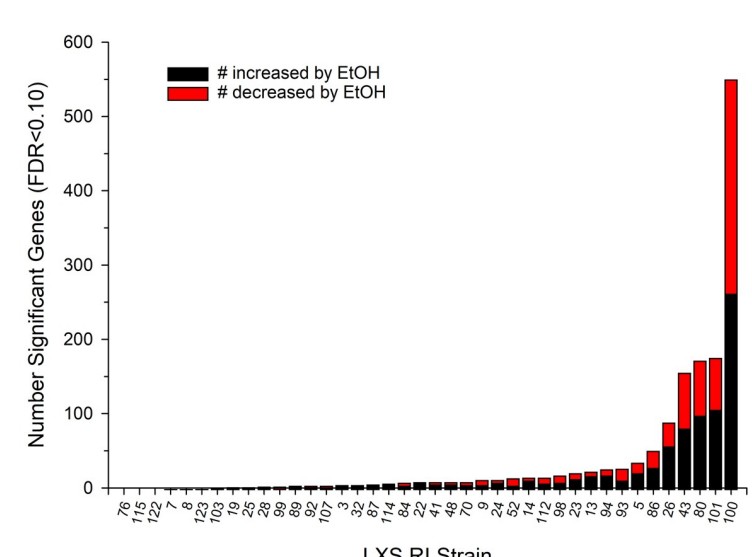

**Fig 2. Genes that were differentially expressed (DE) due to ethanol (EtOH) pretreatment in the LXS RI panel (FDR<0.05).** (A) Number of DE genes by RI strain. Strains 76, 115, and 122 did not have any DE genes. (B) Distribution of the fold-change in expression due to ethanol pretreatment (log2; n=1,230).

We compared the DE genes from our analysis to genes from 21 other published studies that made comparisons between a control condition and some kind of exposure to ethanol. Twelve of the studies were conducted in mouse brain and the remaining 9 were conducted in post-mortem brain tissue from chronic alcoholics (a brief description of the studies can be found in S2 Table). Approximately 72% of our DE genes (887/1,230) were identified as DE in at least one of the studies. The most frequently detected genes from this analysis are listed in Table 1; *i.e.*, the 29 genes that were found in 6 or more of the studies.

Expression QTLs (eQTLs) were mapped using QTL Reaper. Here we report only significant eQTLs; *i.e.*, genome-wide $p < 0.05$ determined by permutation testing for each individual gene (complete mapping results can be found in S3 Table). There were 1,989 and 1,081 *cis*- and *trans*-eQTLs, respectively, in the saline pretreatment group, and 2,016 and 1,328 *cis*- and

**Table 1. LXS DE genes found to be regulated by ethanol in other studies.**

| Symbol | Name[1] | # LXS Strains[2] | # Studies (mouse, human)[3] | References[4] |
|--------|---------|------------------|------------------------------|----------------|
| *Fkbp5* | FK506 binding protein 5 | 5 | 9 (7, 2) | [41–49] |
| *Cdkn1a* | Cyclin-dependent kinase inhibitor 1A (P21) | 5 | 6 (5, 1) | [43–47, 49] |
| *Kcnq2* | Potassium voltage-gated channel, subfamily Q, member 2 | 5 | 6 (5, 1) | [43, 45–47, 49, 50] |
| *Tef* | Thyrotroph embryonic factor | 5 | 6 (6, 0) | [43, 45, 47, 49–51] |
| *Camk1g* | Calcium/calmodulin-dependent protein kinase I gamma | 4 | 6 (5, 1) | [44–47, 49, 50] |
| *Sult1a1* | Sulfotransferase family 1A, phenol-preferring, member 1 | 3 | 7 (7,0) | [43, 45, 47, 49–52] |
| *Pdk4* | Pyruvate dehydrogenase kinase, isoenzyme 2 | 3 | 6 (4, 2) | [42, 43, 45–47, 49] |
| *Rhou* | Ras homolog family member U | 3 | 6 (6, 0) | [41, 43, 44, 47, 49, 50] |
| *Tsc22d3* | TSC22 domain family, member 3 | 2 | 9 (7, 2) | [42, 43, 45–47, 49–52] |
| *Agt* | Angiotensinogen (serpin peptidase inhibitor, clade A, member 8) | 2 | 7 (3, 4) | [45, 46, 49, 50, 53–55] |
| *Zbtb16* | Zinc finger and BTB domain containing 16 | 2 | 7 (5,2) | [42, 43, 46, 47, 49–51] |
| *Fgfrl1* | Fibroblast growth factor receptor-like 1 | 2 | 6 (5,1) | [43–47, 52] |
| *Jun* | Jun proto-oncogene | 2 | 6 (6, 0) | [41, 43, 45, 47, 49, 50] |
| *Utp6* | UTP6 small subunit processome component | 2 | 6 (5, 1) | [42, 43, 45, 47, 49, 50] |
| *Chordc1* | Cysteine and histidine-rich domain (CHORD)-containing, zinc-binding protein 1 | 1 | 7 (5,2) | [42–47, 51] |
| *Galnt9* | Polypeptide N-acetylgalactosaminyltransferase 9 | 1 | 7 (6, 1) | [43–47, 49, 50] |
| *Mt2* | Metallothionein 2 | 1 | 7 (7, 0) | [41, 43, 45, 47, 49, 50, 56] |
| *Cacna1g* | Calcium channel, voltage-dependent, T type, alpha 1G subunit | 1 | 6 (5, 1) | [44–46, 49, 50, 52] |
| *Gabbr1* | Gamma-aminobutyric acid (GABA) B receptor, 1 | 1 | 6 (4, 2) | [43, 49–51, 55, 57] |
| *Gas7* | Growth arrest specific 7 | 1 | 6 (4, 2) | [45–47, 49, 50, 57] |
| *Kif1b* | Kinesin family member 1B | 1 | 6 (5, 1) | [44–47, 49, 50] |
| *Mtdh* | Metadherin | 1 | 6 (5, 1) | [44–47, 49, 50] |
| *Pltp* | Phospholipid transfer protein | 1 | 6 (5, 1) | [46, 47, 49, 50, 56, 58] |
| *S100a10* | S100 calcium binding protein A10 (calpactin) | 1 | 6 (4, 2) | [46, 47, 49, 50, 57, 58] |
| *S100b* | S100 protein, beta polypeptide, neural | 1 | 6 (6, 0) | [43–45, 47, 49, 50] |
| *Sdc4* | Syndecan 4 | 1 | 6 (5, 1) | [42, 43, 45, 47, 49, 50] |
| *Sez6* | Seizure related gene 6 | 1 | 6 (5, 1) | [41, 43, 44, 46, 49, 50] |
| *Sgk1* | Serum/glucocorticoid regulated kinase 1 | 1 | 6 (6, 0) | [43–45, 47, 49, 50] |
| *Tardbp* | TAR DNA binding protein | 1 | 6 (4, 2) | [42, 44–47, 49] |

[1] From MGI (http://www.informatics.jax.org).

[2] Number of LXS RI strains that that showed a significant effect of ethanol pretreatment.

[3] Number of studies that showed an effect of ethanol. Numbers in parentheses indicate how many of the studies were conducted in mouse or human. A total of 12 mouse studies and 9 human studies were examined.

[4] Brief descriptions of the studies can be found in S2 Table.

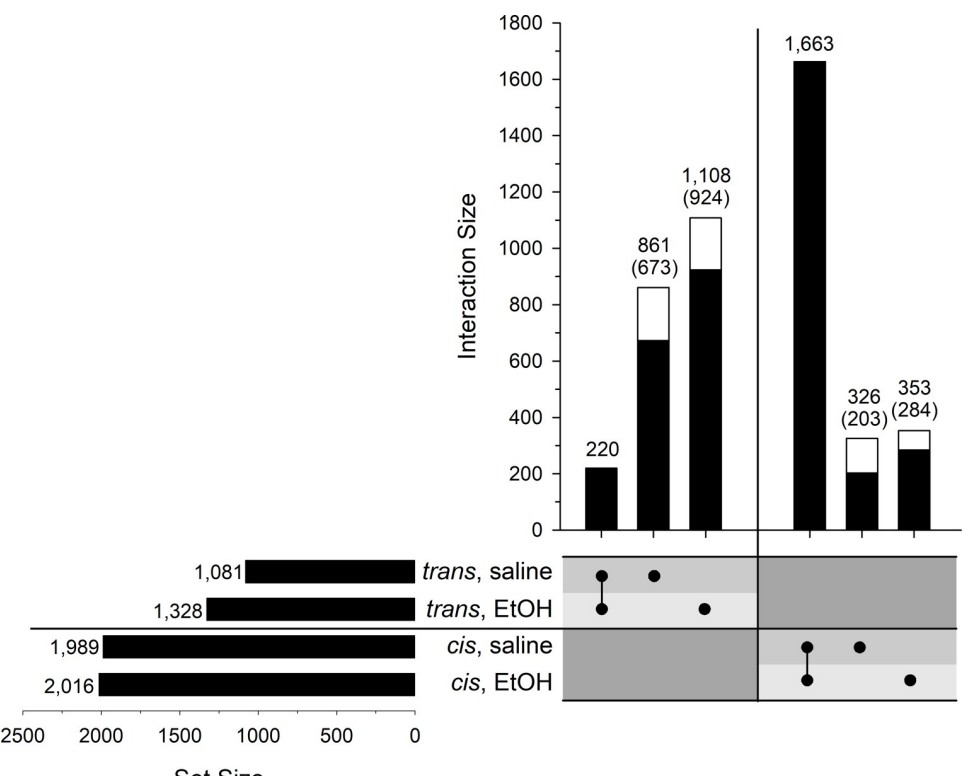

**Fig 3. UpSet plot of the number of eQTLs in the LXS RI panel following saline or ethanol (EtOH) pretreatment.**
Set Size is the total number of eQTLs in each group. Interaction Size represents the number of eQTLs in the
intersections of the groups as indicated by the black dots below the bars. The black portion of the bars in the
Interaction panel represents the number of pretreatment-unique eQTLs that are "golden" (see text). The number of
those "golden" eQTLs is shown in parentheses.

*trans*-eQTLs, respectively, in the ethanol pretreatment group (Fig 3). The majority of *cis*-eQTLs were in common between the two pretreatment groups; however, there was a substantial number of *cis*-eQTLs that were pretreatment-unique (Fig 3). Uniqueness was defined simply as being significant in one pretreatment group (genome-wide p<0.05), but not the other. Similarly, there were a large number of pretreatment-specific *trans*-eQTLs, although it is clear that there was proportionally more unique *trans*-eQTLs than unique *cis*-eQTLs (Fig 3).

A common method of distinguishing *cis*- from *trans*-regulation is to define a distance between the gene location and the location of the eQTL peak. This number, here defined as 5 Mb, tends to be arbitrary. This potentially caused some number of eQTLs to have been misclassified with regard to the nature of their regulation because their significance peak was located just beyond the 5 Mb threshold. Additionally, there were some number of false-negative eQTLs for which the LRS peak fell just below the significance cutoff. These considerations have implications for the identification of eQTLs that were unique to one or the other pretreatment groups which is important for understanding the genetics of gene expression following acute ethanol. We therefore sought to identify "golden" high-confidence pretreatment-unique *cis*- and *trans*-eQTLs. The numbers in parentheses in Fig 3 indicate the number of golden eQTLs that were identified using the filtering procedure described in the Methods. Examples of common and unique (golden) *trans*- and *cis*-eQTLs are illustrated in S1 Fig. As others have noted (*e.g.*, [59]) and as is evident among the examples, *trans*-eQTLs tend to have lower peak LRS scores than *cis*-eQTLs. The mean LRS scores for all significant *cis*- and *trans*-eQTLs in the

saline group were 49.1 (± 0.7) and 29.0 (± 0.8), respectively. These values were similar in the ethanol pretreatment group: 48.5 (± 0.7) for *cis*-eQTLs and 27.1 (± 0.7) for *trans*-eQTLs. Note that the examples shown in S1C and S1D Fig are unusual in that the genes had both a pretreatment-specific *cis*- and *trans*-eQTL. Most (92%) of the golden genes had only a single significant *cis*- or *trans*-eQTL that was pretreatment-specific.

We previously conducted a genetic correlation analysis between gene expression and LORR-related measures in the LXS RI strains; however, the expression dataset was generated using a microarray platform, not RNA-seq, and the mice were completely naïve [22]. Here we have duplicated the previous analysis using the current transcriptome data which carries all of the advantages of the RNA-seq platform [60] and is more appropriate in that the mice were pretreated with saline or ethanol exactly as we had done for the behavioral experiment [21]. Hundreds of genes were found to be significantly correlated to the behaviors (FDR<0.10) and, as illustrated in Fig 4, there was a substantially greater number of genes that were correlated to ST or AFT in the ethanol pretreatment group compared to the saline group (complete results can be found in S3 Table). There were very few genes in common among the four groups, an observation that was true even when relaxing the statistical criteria (not shown). We also conducted a correlation analysis between the behavioral responses and the number of DE genes within each strain. Significant correlations were observed for ST-Et (*r*=0.39, nominal p=0.01) and AFT-Et (*r*=0.33, nominal p=0.04), but not for ST-sal or AFT-sal (nominal p>0.20).

Not all LXS *cis*-eQTLs would be expected to be regulating the ST phenotype for which the ILS and ISS were selected; *i.e.*, it is likely that some unknown number of eQTLs that have no influence on LORR-related phenotypes became fixed in the ILS and ISS by chance. Variants that modulate gene expression and that do contribute to the behavioral phenotypes should not only be *cis*-regulated, but also correlated to the behavior. There were 15 genes of this type in the saline pretreatment group and 69 genes in the ethanol pretreatment group; the genes are listed in Tables 2–4. As noted in the tables, some of these genes were located within previously published LXS LORR-related behavioral QTLs [22].

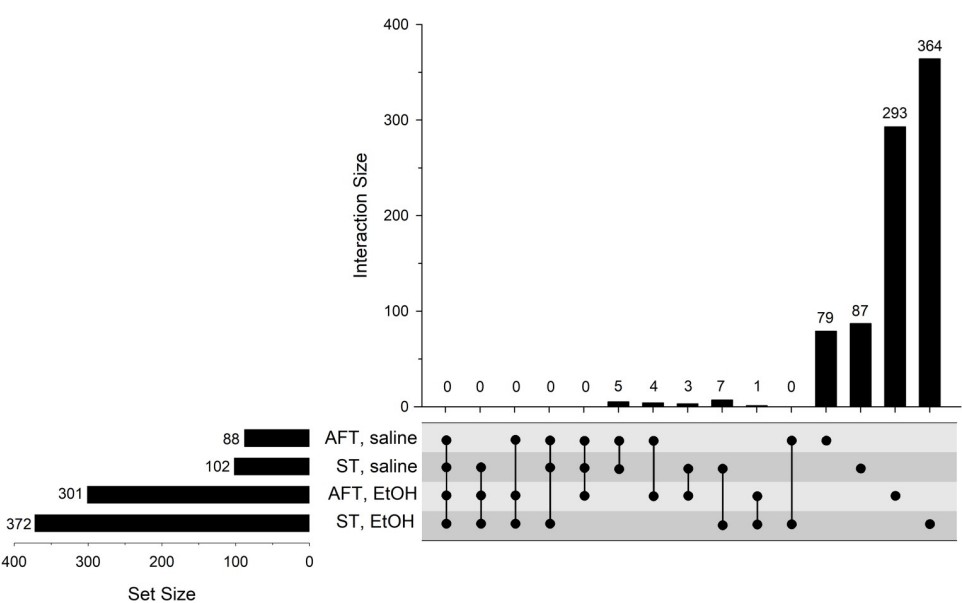

**Fig 4. UpSet plot of the number of genes correlated to AFT or ST in the LXS RI panel following saline or ethanol (EtOH) pretreatment.** See Fig 3 legend for additional details.

**Table 2. Genes that had a significant *cis*-eQTL (genome-wide P<0.05) and were significantly correlated to ST or AFT after saline pretreatment (FDR<0.10).**

| Behavior[1] | Symbol[2] | Chr | Position (Mb)[3] | Behavioral QTL: Chr, peak LRS (CI; Mb)[4] | Name[5] |
|---|---|---|---|---|---|
| ST, saline | *Ankzf1* | 1 | 75.2 | | Ankyrin repeat and zinc finger domain containing 1 |
| | *Chil1* | 1 | 134.2 | 1, 109.2 (65.1 to 135.8) | Chitinase-like 1 |
| | *Adora1* | 1 | 134.2 | 1, 109.2 (65.1 to 135.8) | Adenosine A1 receptor |
| | *Rnpep* | 1 | 135.3 | 1, 109.2 (65.1 to 135.8) | Arginyl aminopeptidase (aminopeptidase B) |
| | *Nav1* | 1 | 135.4 | 1, 109.2 (65.1 to 135.8) | Neuron navigator 1 |
| | *Csrp1* | 1 | 135.7 | 1, 109.2 (65.1 to 135.8) | Cysteine and glycine-rich protein 1 |
| | *Extl1* * | 4 | 134.4 | | Exostoses (multiple)-like 1 |
| | *Chrna6* | 8 | 27.4 | 8, 28.6 (17.4 to 126.9) | Cholinergic receptor, nicotinic, alpha polypeptide 6 |
| | *Sod2* | 17 | 13.0 | | Superoxide dismutase 2, mitochondrial |
| | *Stk19* | 17 | 34.8 | | Serine/threonine kinase 19 |
| | *Tubb5* | 17 | 35.8 | | Tubulin, beta 5 class I |
| AFT, saline | *Dlx1* | 2 | 71.5 | | Distal-less homeobox 1 |
| | *4632427E13Rik* | 7 | 92.7 | | RIKEN cDNA 4632427E13 gene |
| | *Ppp1r9b* | 11 | 95.0 | | Protein phosphatase 1, regulatory subunit 9B |
| | *Ablim1* | 19 | 57.0 | | Actin-binding LIM protein 1 |

[1] Correlated behavior.

[2] Asterisk (*) indicates that the *cis*-eQTL was pretreatment-unique for this gene ("golden").

[3] Gene start position, GRCm38/mm10; from MGI (http://www.informatics.jax.org/).

[4] Indicates whether the gene is located within a QTL for the indicated behavior. Asterisk (*) next to chromosome number indicates that the QTL was significant; all others were suggestive (QTL results from [22]). CI: Bayesian credible interval.

[5] From MGI (http://www.informatics.jax.org/).

We conducted a Gene Ontology (GO) enrichment analysis on the lists of correlated genes and the DE genes using the online tool DAVID (https://david.ncifcrf.gov/; [37]). Genes correlated to ST-sal were enriched for 11 GO *Biological Process* terms, all of which were related to proteolysis, and genes correlated to AFT-sal were enriched for only three terms, all related to gene expression (FDR<0.05; see S4 and S6 Tables). ST-Et and AFT-Et were significantly enriched for a large number of GO *Biological Process* terms, summarized with REVIGO plots in Fig 5A and 5B (FDR<0.05). A similar analysis revealed a large number of significantly enriched GO *Biological Process* terms for the DE gene list (Fig 5C; FDR<0.05). Complete results for the GO *Biological Process* and GO *Cellular Component* enrichment analyses can be found in S4, S5, S6, S7 and S8 Tables. The DE genes were found to significantly overlap with all three cell-specific signatures available in GSEA (neurons, astrocytes, and oligodendrocytes; FDR<0.05); however, none of the correlated gene lists were found to overlap with the expression signatures of these cell types.

Table 5 shows PCLs and their associated perturbagens from the CLUE analysis (τ > |95|). There were many other highly connected individual perturbagens that did not fall into a PCL (complete output can be found in S10, S11, S12, S13, and S14 Tables). Two of the PCL connectivity scores were negative indicating that the perturbagens show an expression profile opposite that of the input profile. Note that the AFT-Et and DE profiles were highly connected to the PCL "HDAC inhibitor" (HDACi). There were 17 and 18 unique HDACi that had profiles similar to the input lists of AFT-Et and DE genes, respectively.

We next explored *Hdac1* (histone deacetylase 1) for its possible role in modulation of AFT after an ethanol pretreatment. We first identified 437 genes whose expression was correlated to *Hdac1* expression in the ethanol pretreatment group (nominal p<0.01; S3 Table). The locations of these genes are illustrated in the circos plot shown in Fig 6. Also shown are the

**Table 3. Genes that had a significant *cis*-eQTL (genome-wide P<0.05) and were significantly correlated to ST after ethanol pretreatment (FDR<0.10).**

| Behavior[1] | Symbol[2] | Chr | Position (Mb)[3] | Behavioral QTL: Chr, peak LRS (CI; Mb)[4] | Name[5] |
|---|---|---|---|---|---|
| ST, EtOH | *Chst10* | 1 | 38.9 | | Carbohydrate sulfotransferase 10 |
| | *Nck2* * | 1 | 43.4 | 1, 80.6 (41.7 to 135.8) | Non-catalytic region of tyrosine kinase adaptor protein 2 |
| | *Slc39a10* | 1 | 46.8 | 1, 80.6 (41.7 to 135.8) | Solute carrier family 39 (zinc transporter), member 10 |
| | *Sox13* | 1 | 133.4 | 1, 80.6 (41.7 to 135.8) | SRY (sex determining region Y)-box 13 |
| | *Adora1* | 1 | 134.2 | 1, 80.6 (41.7 to 135.8) | Adenosine A1 receptor |
| | *Ppp1r12b* | 1 | 134.8 | 1, 80.6 (41.7 to 135.8) | Protein phosphatase 1, regulatory subunit 12B |
| | *Kcna3* | 3 | 107.0 | | Potassium voltage-gated channel, shaker-related subfamily, member 3 |
| | *Actr3b* * | 5 | 25.8 | | ARP3 actin-related protein 3B |
| | *Grk4* | 5 | 34.7 | | G protein-coupled receptor kinase 4 |
| | *Smim14* | 5 | 65.4 | | Small integral membrane protein 14 |
| | *Slain2* | 5 | 72.9 | | SLAIN motif family, member 2 |
| | *Cyth3* * | 5 | 143.6 | | Cytohesin 3 |
| | *Fkbp4* | 6 | 128.4 | | FK506 binding protein 4 |
| | *Far2* | 6 | 148.0 | | Fatty acyl CoA reductase 2 |
| | *Sdhaf1* * | 7 | 30.3 | | Succinate dehydrogenase complex assembly factor 1 |
| | *Ptpn5* | 7 | 47.1 | | Protein tyrosine phosphatase, non-receptor type 5 |
| | *Synm* | 7 | 67.7 | | Synemin, intermediate filament protein |
| | *Relt* | 7 | 100.8 | | RELT tumor necrosis factor receptor |
| | *Agpat5* | 8 | 18.8 | 8, 28.6 (18.3 to 33.1) | 1-acylglycerol-3-phosphate O-acyltransferase 5 (lysophosphatidic acid acyltransferase, epsilon) |
| | *Plpbp* | 8 | 27.0 | 8, 28.6 (18.3 to 33.1) | pyridoxal phosphate binding protein |
| | *Eif4ebp1* | 8 | 27.3 | 8, 28.6 (18.3 to 33.1) | Eukaryotic translation initiation factor 4E binding protein 1 |
| | *Wwc2* | 8 | 47.8 | | WW, C2 and coiled-coil domain containing 2 |
| | *Adi1* | 12 | 28.7 | 12, 34.6 (16.3 to 53.5) | Acireductone dioxygenase 1 |
| | *Dock4* | 12 | 40.4 | 12, 34.6 (16.3 to 53.5) | Dedicator of cytokinesis 4 |
| | *Dtd2* | 12 | 52.0 | 12, 34.6 (16.3 to 53.5) | D-tyrosyl-tRNA deacylase 2 |
| | *Dhrs7* | 12 | 72.6 | | Dehydrogenase/reductase (SDR family) member 7 |
| | *Acat2* | 17 | 12.9 | 17, 35.8 (9.3 to 47.4) | Acetyl-Coenzyme A acetyltransferase 2 |
| | *Atp6v1g2* | 17 | 35.2 | 17, 35.8 (9.3 to 47.4) | ATPase, H+ transporting, lysosomal V1 subunit G2 |
| | *Tubb5* | 17 | 35.8 | 17, 35.8 (9.3 to 47.4) | Tubulin, beta 5 class I |

[1] Correlated behavior.

[2] Asterisk (*) indicates that the *cis*-eQTL was pretreatment-unique for this gene ("golden").

[3] Gene start position, GRCm38/mm10; from MGI (http://www.informatics.jax.org/).

[4] Indicates whether the gene is located within a QTL for the indicated behavior. Asterisk (*) next to chromosome number indicates that the QTL was significant; all others were suggestive (QTL results from [22]). CI: Bayesian credible interval.

[5] From MGI (http://www.informatics.jax.org/).

absolute values of the genetic correlation coefficients (*r*) between these genes and AFT-Et. There is an obvious cluster of genes in close proximity to *Hdac1* and it is possible that some number of them were correlated due to linkage disequilibrium (LD) effects and not because of regulation by *Hdac1*. It is thus notable that 74% of the genes (332) were on different chromosomes or were more than 20 Mb away from *Hdac1* on chromosome 4, a distance that makes LD effects unlikely. It is also notable that 58% of the genes (252) were correlated to AFT-Et at a nominal value of p<0.05.

A DAVID enrichment analysis indicated that the 437 genes that were correlated to *Hdac1* were significantly enriched for 25 GO Biological Process terms (FDR<0.05; see S9 Table for

**Table 4. Genes that had a significant *cis*-eQTL (genome-wide P<0.05) and were significantly correlated to AFT after ethanol pretreatment (FDR<0.10).**

| Behavior[1] | Symbol[2] | Chr | Position (Mb)[3] | Behavioral QTL: Chr, peak LRS (CI, Mb)[4] | Name[5] |
|---|---|---|---|---|---|
| AFT, EtOH | Cyp2j12 | 4 | 96.1 | | Cytochrome P450, family 2, subfamily j, polypeptide 12 |
| | Atg4c | 4 | 99.2 | | Autophagy related 4C, cysteine peptidase |
| | Alg6 | 4 | 99.7 | | Asparagine-linked glycosylation 6 (alpha-1,3,-glucosyltransferase) |
| | Efcab7 | 4 | 99.8 | | EF-hand calcium binding domain 7 |
| | Mier1 | 4 | 103.1 | | MEIR1 transcription regulator |
| | Lrp8 | 4 | 107.8 | | Low density lipoprotein receptor-related protein 8, apolipoprotein e receptor |
| | Cyp4x1 | 4 | 115.1 | | Cytochrome P450, family 4, subfamily x, polypeptide 1 |
| | Ppih | 4 | 119.3 | 4*, 129.8 (116.0 to 139.1) | Peptidyl prolyl isomerase H |
| | Ago3 | 4 | 126.3 | 4*, 129.8 (116.0 to 139.1) | Argonaute RISC catalytic subunit 3 |
| | CK137956 | 4 | 127.9 | 4*, 129.8 (116.0 to 139.1) | cDNA sequence CK137956 |
| | Hdac1 | 4 | 129.5 | 4*, 129.8 (116.0 to 139.1) | Histone deacetylase 1 |
| | Iqcc | 4 | 129.6 | 4*, 129.8 (116.0 to 139.1) | IQ motif containing C |
| | 1700003M07Rik | 4 | 130.0 | 4*, 129.8 (116.0 to 139.1) | RIKEN cDNA 1700003M07 gene |
| | Col16a1 | 4 | 130.0 | 4*, 129.8 (116.0 to 139.1) | Collagen, type XVI, alpha 1 |
| | Hmgcl | 4 | 135.9 | 4*, 129.8 (116.0 to 139.1) | 3-hydroxy-3-methylglutaryl-Coenzyme A lyase |
| | Asap3 | 4 | 136.2 | 4*, 129.8 (116.0 to 139.1) | ArfGAP with SH3 domain, ankyrin repeat and PH domain 3 |
| | Luzp1 | 4 | 136.5 | 4*, 129.8 (116.0 to 139.1) | Leucine zipper protein 1 |
| | Ece1 | 4 | 137.9 | 4*, 129.8 (116.0 to 139.1) | Endothelin converting enzyme 1 |
| | Htr6 * | 4 | 139.1 | 4*, 129.8 (116.0 to 139.1) | 5-hydroxytryptamine (serotonin) receptor 6 |
| | Cep135 | 5 | 76.6 | | Centrosomal protein 135 |
| | Naaa | 5 | 92.3 | | N-acylethanolamine acid amidase |
| | Sept11 * | 5 | 93.1 | | Septin 11 |
| | Mvk * | 5 | 114.4 | | Mevalonate kinase |
| | Chrm2 * | 6 | 36.4 | | Cholinergic receptor, muscarinic 2, cardiac |
| | Smg9 | 7 | 24.4 | | SMG-9 homolog, nonsense mediated mRNA decay factor (*C. elegans*) |
| | Dmac2 | 7 | 25.6 | | Distal membrane arm assembly complex 2 |
| | Fbxo27 * | 7 | 28.7 | | F-box protein 27 |
| | Fxyd5 | 7 | 31.0 | | FXYD domain-containing ion transport regulator 5 |
| | Rras2 | 7 | 114.0 | | Related RAS viral (r-ras) oncogene 2 |
| | Usp31 * | 7 | 121.6 | | Ubiquitin specific peptidase 31 |
| | Kdm8 | 7 | 125.4 | | Lysine (K)-specific demethylase 8 |
| | Elmod1 * | 9 | 53.9 | | ELMO/CED-12 domain containing 1 |
| | Cmtm6 | 9 | 114.7 | | CKLF-like MARVEL transmembrane domain containing 6 |
| | Map3k5 | 10 | 19.9 | | Mitogen-activated protein kinase kinase kinase 5 |
| | Nup107 | 10 | 117.8 | | Nucleoporin 107 |
| | Coil | 11 | 89.0 | | Coilin |
| | Rbm26 * | 14 | 105.1 | | RNA binding motif protein 26 |
| | Adamts1 | 16 | 85.8 | | A disintegrin-like and metallopeptidase (reprolysin type) with thrombospondin type 1 motif, 1 |
| | Eml4 | 17 | 83.4 | | Echinoderm microtubule associated protein like 4 |
| | Fam204a | 19 | 60.2 | | Family with sequence similarity 204, member A |

[1] Correlated behavior.

[2] Asterisk (*) indicates that the *cis*-eQTL was pretreatment-unique for this gene ("golden").

[3] Gene start position, GRCm38/mm10; from MGI (http://www.informatics.jax.org/).

[4] Indicates whether the gene is located within a QTL for the indicated behavior. Asterisk (*) next to chromosome number indicates that the QTL was significant; all others were suggestive (QTL results from [22]). CI: Bayesian credible interval.

[5] From MGI (http://www.informatics.jax.org/).

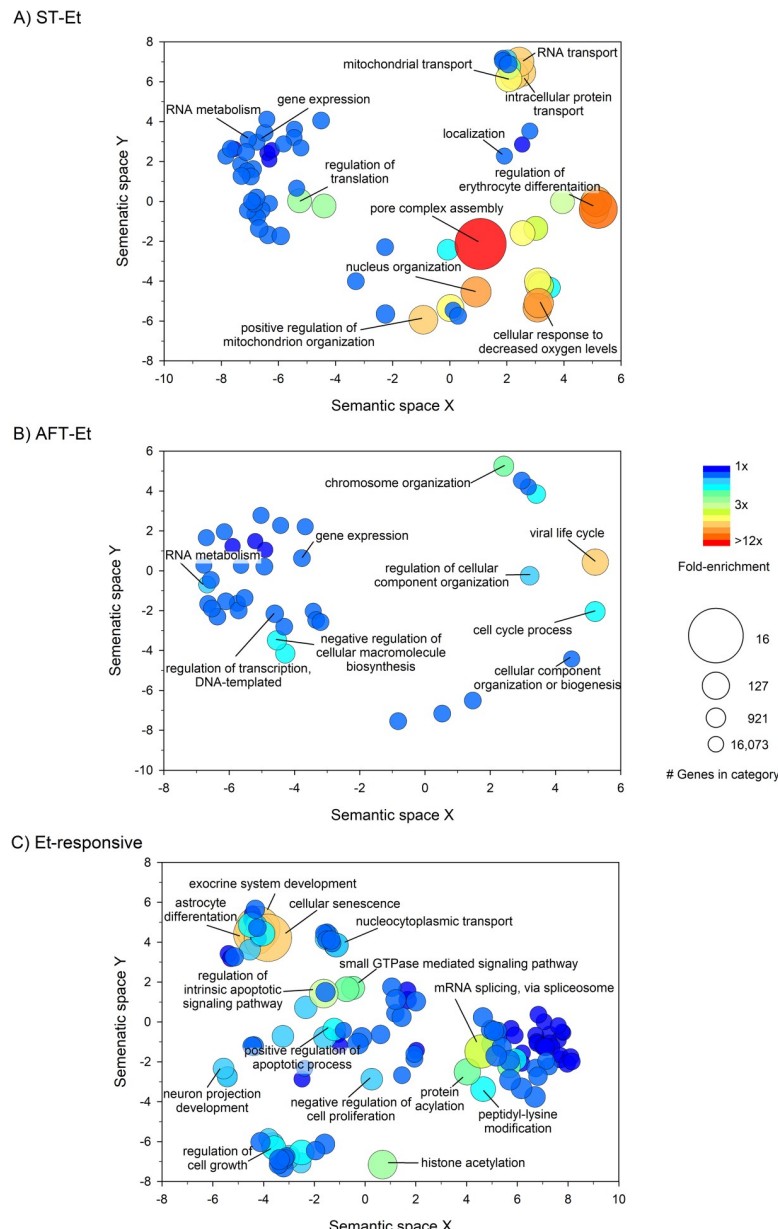

**Fig 5. REVIGO plots of significantly enriched GO *Biological Process* terms.** (A) Genes correlated to ST, ethanol (Et) pretreatment. (B) Genes correlated to AFT, ethanol (Et) pretreatment. (C) DE genes. Fold enrichment is the proportion of term genes found in the input list compared to the proportion of total term genes found in the background. The size of the bubble is inversely proportional to the number of genes in the term; *i.e.*, the larger the bubble, the more specific the term.

full results). REVIGO output for these terms is shown in S2 Fig. Two terms in particular stood out: "neuron-neuron synaptic transmission" (10 genes) and "potassium ion transport" (13 genes). The former included three glutamatergic receptors or receptor subunits (*Grm2*, *Grik3*, and *Grin2d*) and the latter included a calcium-activated potassium channel (*Kcnn3*) and a voltage-gated potassium channel (*Kcnh7*). All of these genes were correlated to AFT-Et (nominal p<0.05) with the exception of *Grm2* which did not quite reach this level of significance (*r*=-

**Table 5. CMap perturbagen classes for the correlated gene lists and the DE gene list with a summary connectivity score greater than |95|.**

| Gene list (# genes)[1] | Perturbagen class (PCL) | Summary connectivity score (τ)[2] | # Perturbagens[3] |
|---|---|---|---|
| ST-sal, correlated genes (166) | Protein synthesis inhibitor | -98.2 | 5 |
| ST-Et, correlated genes (229) | (none) | (NA) | (NA) |
| AFT-sal, correlated genes (54) | (none) | (NA) | (NA) |
| AFT-Et, correlated genes (209) | HDAC inhibitor | 99.7 | 17 |
| | Leucine rich repeat kinase inhibitor | 99.0 | 2 |
| | Bromodomain Inhibitor | 98.0 | 1 |
| | PKC activator | 96.9 | 3 |
| DE genes (300) | HDAC inhibitor | 99.6 | 18 |
| | DNA dependent protein kinase inhibitor | 98.1 | 2 |
| | FGFR inhibitor | -97.8 | 2 |
| | Bromodomain inhibitor | 96.4 | 1 |
| | CDK inhibitor | 95.9 | 8 |

[1] The CLUE query input is restricted to a total of 300 genes. Only genes that had a valid HUGO symbol and were part of BING space were used in the query.

[2] Only shown are PCLs with a connectivity score (τ) > |95|.

[3] Number of individual perturbagens that were within the PCL including perturbagens that were not considered to be part of the core PCL, but that had the PCL name in their description or in their MOA. Connectivity score (τ) > |90| for the individual perturbagens.

0.27, nominal p=0.09). Another interesting term was "regulation of RNA splicing" (9 genes), a term for which the DE genes were also enriched.

## Discussion

In our ongoing efforts to understand the basis of genetic variation in LORR-related responses and its relationship to drinking behavior, we have used RNA-seq to profile the brain transcriptome of the LXS RI panel 8 hours after being administered saline or ethanol. The time point was chosen based on previous studies [22] and the ethanol dose was identical to that used in parallel behavioral studies in the LXS [21, 22]. The design allowed for a comparison between gene expression and LORR-related behaviors in the two pretreatment groups and for the examination of the influence of genetics on ethanol-mediated modulation of gene expression. The results indicate prominent genetic and pretreatment effects on the regulation of gene expression with important strain-by-pretreatment (gene-by-environment; GxE) interaction effects. Moreover, the results suggest some specific genes and pathways that may be regulating ST or AFT.

Expression abundance of more than a third of the genes analyzed was significantly heritable suggesting widespread genetic regulation of gene expression in the mouse brain as has been noted by others (*e.g.*, [61]). Interestingly, there was a greater number of genes with significant heritability in the ethanol pretreatment group than in the saline group, and heritabilities for genes from the ethanol group tended to be higher than the saline group. Heritability in RI strains is fundamentally a function of between- and within-strain variance ($V_B$; $V_W$); an increase in the former and/or a decrease in the latter increases heritability. In fact, we observed both effects: $V_B$ tended to be slightly higher and $V_W$ lower in the ethanol pretreatment group when examined across all 14,184 expressed genes (not shown). The increased $V_B$ could be due to a broadening of the strain distribution for given gene due to ethanol-induced increases or decreases in the gene's expression in one or more strains. It is unclear, however, why the ethanol pretreatment would cause a decrease in $V_W$. Whatever the underlying cause, the sheer large number of genes that were significantly heritable may be indicating that a substantial

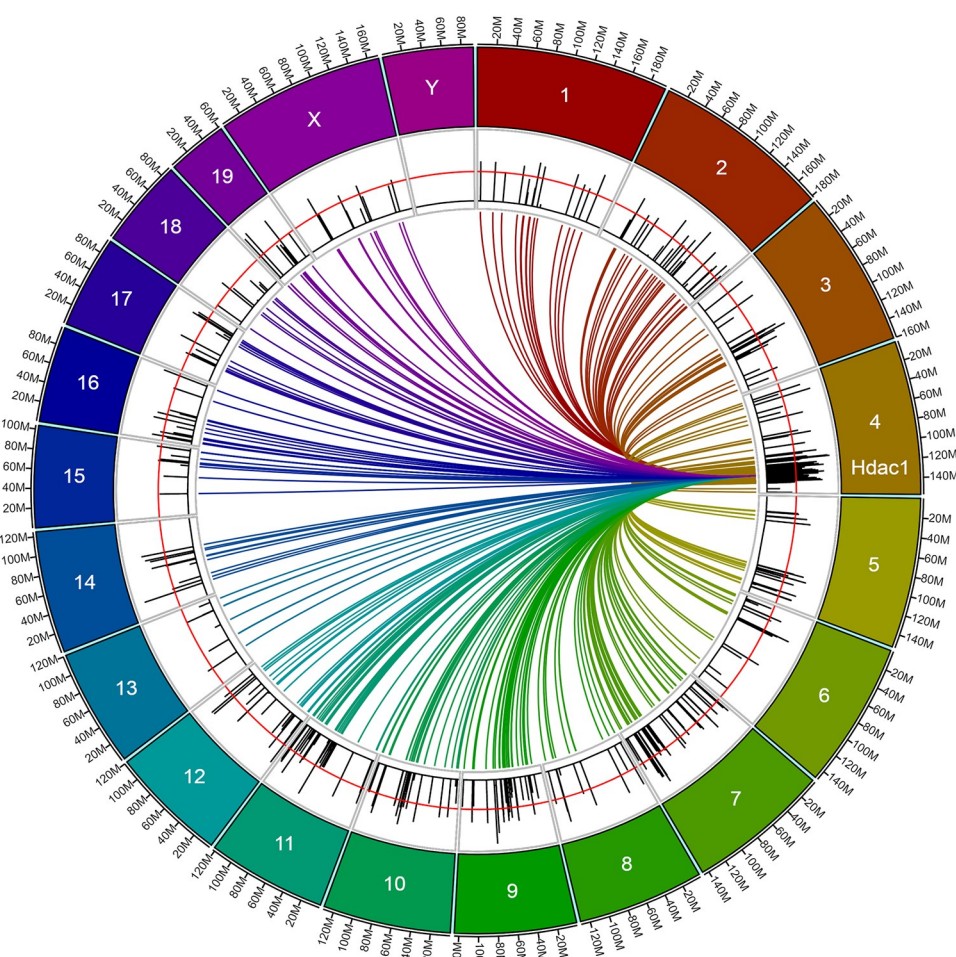

**Fig 6. Circos plot of genes correlated to *Hdac1* in the ethanol pretreatment group (nominal p<0.01; n=446).** Colored lines connect *Hdac1* to the location of the genes to which it is correlated. Black bars for each gene show the absolute value of the Pearson product-moment correlation coefficient (*r*) between the gene's expression and AFT-Et (range of *r* values: <0.01 to 0.67). The red line over the black bars indicates the nominal p<0.05 cutoff for the correlation between gene expression and AFT-Et (*r* ≅ 0.31).

number of genes likely contributes to genetic variation in the LORR phenotypes, although certainly not all do.

The results indicate that acute ethanol has a broad, pronounced effect on gene expression in the mammalian brain, an observation in line with other studies (*e.g.*, [41]). There also was a clear genetic effect on the regulation of gene expression by acute ethanol; *i.e.*, the number of significant DE genes varied considerably by strain, from as few as 0 in several strains to over 500 in the most sensitive strain. Consistent with the two-way ANOVA, this result indicates prominent GxE interactions. Others have reported strain-by-ethanol interactions on gene expression following acute (*e.g.*, [41]) or chronic (*e.g.*, [56]) administration, but the range and magnitude of the effect observed here demonstrates the importance that genetic makeup can have on an individual's transcriptome response to acute ethanol.

Ethanol-mediated effects on the transcriptome vary based on dose, method of administration, length of exposure, and brain region [62]. The method employed here was used to identify genes that were correlated to specific behaviors that were tested under identical treatment conditions [21] and would not necessarily be expected to be the same as those that change

under other conditions of ethanol exposure, including in humans following many years of drinking. It is thus interesting that 63% (769) and 37% (460) of our DE genes were identified in one or more transcriptome studies conducted in mouse brain or human alcoholic brains, respectively. This is remarkable considering how different the methods and procedures were in the studies, and it suggests a high degree of confidence in our results. It also suggests that the overlapping sets of genes are robust markers for exposure to ethanol and may provide clues to the underlying mechanistic basis of the brain's response to ethanol. For example, FK506 binding protein 5 (*Fkbp5*) was the most consistent DE gene across studies. It was up-regulated in 7 of the 9 studies in which it was DE (direction of change was not provided for the other two studies) and it was significantly DE and up-regulated in 5 of the LXS RI strains. *Fkbp5* expression was correlated to ST-Et ($r$ = 0.41, nominal p=0.009) and to AFT-Et ($r$ = -0.32, nominal p=0.047) suggesting that it may be involved in these behaviors. The FKBP5 protein participates in the regulation of glucocorticoid signaling [63] and it has been implicated in various neuropsychiatric disorders including depression, bipolar disorder, and schizophrenia [64]. Stress pathways, including glucocorticoid signaling, have long been implicated in AUDs [65]. Direct evidence for a role for *Fkbp5* comes from Qiu *et al.* [66] who observed increased drinking in an *Fkbp5* null mutant mouse and a genetic association between drinking and *FKBP5* SNPs in a sample of college students. In addition, König *et al.* [67] observed that a selective inhibitor of FKBP51 (the rat orthologue) reduced drinking in a two-bottle choice test in rats. The current observations along with these varied results suggest a role for *Fkbp5* in the ethanol responses tested here and possibly in AUDs.

Functional enrichment analysis of the DE genes revealed several categories that may be contributing to adaptive responses to ethanol such as tolerance or sensitization. For example, "mRNA splicing" was a highly enriched term and altered regulation of alternative splicing has been proposed to contribute to ethanol sensitization in the mouse [43]. It also has been implicated in relevant changes to transcriptome architecture following long-term chronic drinking in humans [68]. A perhaps unsurprising enriched term was apoptosis given the large pretreatment dose of ethanol. Enrichment for apoptosis-related genes has been noted in both rodent and human chronic administration studies [57, 69]. It has been postulated that brain structures involved in executive functions are more sensitive to cell death resulting from high-dose acute (binge drinking) and/or chronic exposure and that this effect may be contributing to the addiction process [70].

A central hypothesis of this project is that genes whose expression co-varies with behavior are in some way involved in the behavior. Such genes in the saline pretreatment group would be considered "predisposing"; *i.e.*, their baseline expression prior to being tested for LORR contributes to variance in the behavioral response. The acute ethanol administered for the LORR test changed the expression of many genes in a strain-dependent manner, as noted above; however, it is unlikely that these genes contributed much if any to the behavioral output for the saline pretreatment group due to the relatively short duration over which LORR occurs. This was the fundamental basis for profiling the transcriptome 8 hours after an ethanol pretreatment and comparing that to LORR responses 24 hours after the ethanol pretreatment: ethanol-mediated alterations in the expression of trait-relevant genes would be expected to be correlated to ethanol-mediated alterations in the behavioral traits. The correlated genes should then provide clues about the underlying molecular mechanisms.

The enrichment analysis of the correlated genes in the saline pretreatment group – the "predisposing" genes – was not especially revealing. It is notable, however, that ST-sal was enriched for the GO *Cellular Component* term "myelin sheath" (see S4 Table). It is well established that ethanol interferes in some way with oligodendrocytes and the myelin sheaths they form around CNS axons, particularly in humans following chronic consumption [71, 72]. Effects on

the expression of myelin-related genes also has been observed following acute exposure in rodents [41]. The finding that myelin-related genes are enriched among genes correlated to ST, including *Mbp* (myelin basic protein), raises the interesting possibility that differences in myelin structure and/or function in naïve mice is a "predisposing" condition that contributes to variation in ST. *Mbp* was not *cis*-regulated, but it was highly heritable in the LXS ($H^2$=0.58) and, of course, significantly correlated to ST-sal ($r = 0.50$, p = 0.001). Two naturally occurring mouse mutations of *Mbp*, shi and mld, show a variety of CNS-related phenotypes including abnormal neurotransmission, ataxia, seizures, and poor learning ability [73–76]. All of these phenotypes could influence ethanol-related behaviors; however, to our knowledge, ethanol has not been tested in the mutant mice.

In contrast to the saline pretreatment, genes correlated to ST or AFT in the ethanol group were significantly enriched for a wide variety of terms. Both ST-Et and AFT-Et were enriched for genes involved in transcription with the former also enriched for translation-related genes. Genetic differences in the processing of RNA combined with ethanol's effects on gene expression could contribute to a strain's specific expression profile and behavioral response. Indeed, the finding that ST and AFT in the ethanol pretreatment group, but not in the saline group, were positively correlated to the number of DE genes in each strain suggests that an increased transcriptomic response to ethanol is at least partly responsible for an increase in the behavioral response. Other terms relate to chromosome organization, movement of macromolecules between cellular compartments, and cellular and nucleus organization. These findings could be indicating that ethanol exposure is affecting many aspects of basal cellular activity which lead to differences in, for example, synaptic structure or other neuron-related functions that affect behavioral responses to ethanol. It is thus possible that some portion of the genetic variance in behavior is due to a broad strain-dependent cellular remodeling or variation in certain basic cellular functions.

The eQTL analysis revealed GxE interaction effects on gene expression as noted above for the analysis of DE genes; *i.e.*, a proportion of the *cis*- and *trans*-eQTLs were mapped only in the presence of one or the other pretreatments. The majority of the *trans*-eQTLs in the LXS were pretreatment-specific (82%) while the majority of the *cis*-eQTLs were common between the two pretreatments (83%). This pattern is similar to an analysis in which eQTLs were mapped in the LXS and the BXD RI panels following one of four treatments that included combinations of acute ethanol and stress [77]. These authors found that the majority of genes whose expression was definitively *trans*-regulated were treatment-specific whereas the majority of *cis*-regulated genes were in common among treatment groups. Similar observations have been made in other organisms under different treatment conditions [78, 79]. This effect is likely due in part to the direct vs. indirect nature of *cis*- vs. *trans*-regulation; *i.e.*, *trans*-regulation involves one or more intermediates that may be under the influence of the treatment condition. There is likely also an effect of how one specifically defines a *cis*- vs. *trans*-eQTL including the fact that some *trans*-eQTLs, which tend to have lower LRS values, will be just under the chosen statistical threshold and therefore not recognized; however, the conservatively defined golden *trans*-eQTLs still made up 66% of the total. Deeper investigation of the specific DNA variants responsible for the pretreatment-specific eQTLs, particularly *cis*-eQTLs, can potentially provide new information on the regulation of gene expression by ethanol, including DNA elements that are sensitive or insensitive to ethanol in the brain.

eQTL mapping in conjunction with genetic correlation analysis between expression and behavioral traits can be used to identify candidate genes; *i.e.*, genes whose expression is *cis*-regulated, genetically correlated to the trait of interest, and occur within a QTL for the trait. Some of the candidates listed in Tables 2 through 4 are also biologically plausible with evidence linking them to ethanol-related responses; *e.g.*, *Chrna6* (cholinergic receptor, nicotinic, alpha

polypeptide 6; [80]); *Adora1* (adenosine A1 receptor; [81]); *Htr6* (5-hydroxytryptamine receptor 6; [82]); and *Ptpn5* (protein tyrosine phosphatase, non-receptor type 5; [83]). Most of the putative candidates, however, have no known relationship to ethanol and are therefore of even greater interest because they may lead to novel mechanisms of AUD risk. For example, developmental effects are probably underappreciated because genetic studies of behavior tend to focus on genes that are active during the particular developmental period under study and that is typically in the adult. *Nav1* (neuron navigator 1) is restricted to the nervous system and appears to be involved in neuronal migration during early stages of development [84]. It has been found to be associated with NMDA receptor complexes, a system well known to be influenced by ethanol [85]. This raises the interesting possibility that *Nav1* is shaping the developing brain in a way that ultimately leads to differential ethanol-mediated behavioral responses in the adult, although the fact that it was expressed in the adult brain suggests that it may be participating in other functions. The regulation of cytoskeleton as it pertains to synapse structure and function occurs through normal development and in response to environmental cues, including ethanol [86, 87]. Several of the genes that are *cis*-regulated and correlated to behavior are involved with cytoskeleton regulation (*e.g.*, *Cep135* and *Asap3*), especially with regard to dendrite structure (*e.g.*, *Nck2* and *Ppp1r9b*). To our knowledge, however, none of these genes has been associated with ethanol responses.

Exposure to ethanol engages epigenetic mechanisms that alter the chromatin landscape in ways that are thought to mediate long-term behaviors, the consequences of which may contribute to pathological drinking behavior [88]. The regulation of chromatin accessibility through the enzymatic addition or removal of acetyl groups from histones is one such mechanism [89–92]. Here, several lines of evidence point to *Hdac1* as controlling some portion of the genetic variation for AFT-Et. *Hdac1*, a class I histone deacetylase known to regulate transcription through deacetylation of lysine residues on all four core histones [93], was *cis*-regulated in both the saline and ethanol pretreatment groups. Its physical location coincides exactly with the peak of the significant AFT-Et QTL that we previously mapped on chromosome 4 (129.5 Mb; [22]) and it was significantly correlated to AFT-Et ($r = 0.54$), but not to AFT-sal or to ST in either pretreatment group ($|r| < 0.30$; nominal $p > 0.05$). The CLUE results indicate that the AFT-Et signature of correlated genes was similar to signatures derived from numerous HDACi; this was not true for ST-Et, ST-sal, or AFT-sal. This result was particularly striking given that the number of HDACi perturbagens was substantially greater than in any other PCL. We further investigated the possibility that specifically *Hdac1* was involved by identifying the genes that were correlated to *Hdac1* and found that a majority of them (252; 58% of total) were also correlated to AFT-Et. Correlation of a gene's expression to *Hdac1* expression supports that *Hdac1* is regulating the gene's expression, although it is certainly not definitive proof. The enrichment analysis of these correlated genes indicates that they are involved in neuronal signaling; most notably glutamate signaling and potassium channel activity, both of which have been implicated in responses to ethanol [94–97]. The knowledge that *Hdac1* may be involved in AFT-Et is of limited utility in understanding ethanol's actions since it presumably has broad effects in the brain and in other tissues. The current results address not only the genetic mechanism of how the brain may be remodeled following ethanol exposure (*cis*-regulation of *Hdac1*), but to the specific components that are being remodeled (genes correlated *Hdac1* expression).

It is important to point out that there were limitations to certain aspects of this study. The lack of anatomical resolution due to the use of whole brain rather than discrete brain regions undoubtedly caused some genes to be missed and did not provide any insight into the contributions and relationships among specific structures that might be involved in the behavioral responses. A second limitation was the use of bulk tissue as opposed to single-nucleus or

single-cell RNA-seq methods which would have provided information about the effects of genetics and ethanol treatment on individual cell types. The GSEA analysis was not especially revealing in this regard either. Finally, only males were examined. Others have noted sex-dependent effects in the response to ethanol and as a function of genetic make-up, and this is likely true for the LXS as well (*e.g.*, [98]).

Here we have used a strategy of expression profiling conducted in parallel with behavioral studies in which the LXS RI mice were treated identically in both arms of the experiment. While there were experimental limitations, as noted above, the design allowed for a direct comparison of the effects of ethanol on gene expression and on behavior. This is one of the primary benefits of using RI strains: results can be analyzed for covariation in cohorts that were tested for different outcomes, or at different times or places [20]. The results highlight some individual genes and pathways that may contribute to LORR-related responses and possibly to drinking behavior. In particular, *Hdac1* emerged as a high priority candidate for the AFT-Et QTL on chromosome 4. It should be noted, however, that within the Bayesian credible interval of the chromosome 4 QTL, there were over 100 cis-eQTLs as well as 150 protein-coding genes with an exonic indel or non-synonymous SNP (current results; [22]). It is thus possible if not likely that other genes within the interval contribute to the QTL with *Hdac1* perhaps having a more prominent effect. Overall, the results provide several avenues of research to pursue with regard to the genetic influence on LORR-related responses and may aid in understanding the relationship between acute sensitivity and tolerance and drinking behavior.

## Supporting information

**S1 Fig. Pretreatment-unique eQTLs and *Hdac1* REVIGO plot.** Examples of eQTLs that are common or unique as a function of saline or alcohol pretreatment in the LXS RI panel.
(PDF)

**S2 Fig. Pretreatment-unique eQTLs and *Hdac1* REVIGO plot.** REVIGO plot of significantly enriched GO Biological Process terms for genes correlated to *Hdac1* (n=446).
(PDF)

**S1 Table. RNA-seq metrics and transcriptome studies.** RNA-seq library and mapping metrics.
(PDF)

**S2 Table. RNA-seq metrics and transcriptome studies.** Mouse and human transcriptome studies that were compared to the LXS Et-responsive gene list.
(PDF)

**S3 Table. Results of key analyses for 14,184 expressed genes.**
(XLSB)

**S4 Table. Gene ontology enrichment analysis.** GO analysis for genes correlated to ST-sal.
(XLSX)

**S5 Table. Gene ontology enrichment analysis.** GO analysis for genes correlated to ST-Et.
(XLSX)

**S6 Table. Gene ontology enrichment analysis.** GO analysis for genes correlated to AFT-sal.
(XLSX)

**S7 Table. Gene ontology enrichment analysis.** GO analysis for genes correlated to AFT-Et.
(XLSX)

**S8 Table. Gene ontology enrichment analysis.** GO analysis for DE genes.
(XLSX)

**S9 Table. Gene ontology enrichment analysis.** GO analysis for genes correlated to *Hdac1*.
(XLSX)

**S10 Table. Cmap enrichment analysis.** Cmap analysis for genes correlated to ST-sal.
(XLSX)

**S11 Table. Cmap enrichment analysis.** Cmap analysis for genes correlated to ST-Et.
(XLSX)

**S12 Table. Cmap enrichment analysis.** Cmap analysis for genes correlated to AFT-sal.
(XLSX)

**S13 Table. Cmap enrichment analysis.** Cmap analysis for genes correlated to AFT-Et.
(XLSX)

**S14 Table. Cmap enrichment analysis.** Cmap analysis for DE genes.
(XLSX)

## Acknowledgments

The authors would like to thank the University of Colorado Anschutz Medical Campus Genomics and Microarray Core for performing the quantitative RNA sequencing and the BioFrontiers Computing Core at the University of Colorado Boulder for providing High Performance Computing resources.

## Author Contributions

**Conceptualization:** Richard A. Radcliffe, Robin Dowell, Katerina Kechris, Laura M. Saba.

**Data curation:** Robin Dowell, Aaron T. Odell, Phillip A. Richmond, Beth Bennett, Colin Larson, Pratyaydipta Rudra, Shi Wen.

**Formal analysis:** Richard A. Radcliffe, Robin Dowell, Aaron T. Odell, Phillip A. Richmond, Pratyaydipta Rudra, Shi Wen.

**Funding acquisition:** Richard A. Radcliffe.

**Investigation:** Beth Bennett, Colin Larson.

**Project administration:** Richard A. Radcliffe.

**Resources:** Richard A. Radcliffe.

**Supervision:** Richard A. Radcliffe, Robin Dowell, Katerina Kechris, Laura M. Saba.

**Visualization:** Richard A. Radcliffe.

**Writing – original draft:** Richard A. Radcliffe.

**Writing – review & editing:** Richard A. Radcliffe, Robin Dowell, Beth Bennett, Katerina Kechris, Laura M. Saba, Pratyaydipta Rudra.

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
