## [Decision Letter · Decision Letter 0]

4 Sep 2020

PONE-D-20-24132

Systems genetics analysis of the LXS recombinant inbred mouse strains: Genetic and molecular insights into acute ethanol tolerance

PLOS ONE

Dear Dr. Richard Radcliffe:

Thank you for submitting your manuscript to PLOS ONE. After careful consideration, we feel that it has merit but does not fully meet PLOS ONE’s publication criteria as it currently stands. Therefore, we invite you to submit a revised version of the manuscript that addresses the points raised during the review process.

We look forward to receiving your revised manuscript.

Kind regards,

Doo-Sup Choi

Academic Editor

PLOS ONE

Journal Requirements:

Reviewers' comments:

Reviewer's Responses to Questions

**Comments to the Author**

1. Is the manuscript technically sound, and do the data support the conclusions?

Reviewer #1: Yes

Reviewer #2: Yes

2. Has the statistical analysis been performed appropriately and rigorously? 

Reviewer #1: I Don't Know

Reviewer #2: Yes

3. Have the authors made all data underlying the findings in their manuscript fully available?

Reviewer #1: No

Reviewer #2: Yes

4. Is the manuscript presented in an intelligible fashion and written in standard English?

Reviewer #1: Yes

Reviewer #2: Yes

5. Review Comments to the Author

Reviewer #1: This study provides new and potentially useful expression profiling of a large group of RI mice with and without alcohol treatment. The authors have done a great job of providing useful and detailed introduction and discussion for their work. I have some concerns about the work and presentation:

1. Some journals now require a section on limitations and caveats of the study and this one certainly needs such a discussion. Two major limitations are the use of whole brain and the small sample size (1, 2 or 3). All brain transcriptome studies show that brain region (and cell types) are major contributors of variance, which are lost here. And what are the implications of having n=1 for some samples?

2. Continuing with sample size, the Discussion states “The number of significant DE genes varied considerably by strain, from as few as 0 in several strains to over 500 in the most sensitive strain.” As I understand the methods, some of these comparisons are based on n=1, others on 2 or 3. How can a comparison with a single sample be statistically significant?

3. There is no mention of cell type except a bit of discussion about myelin. The data could be analyzed for some glial signatures, as has been done in other brain gene expression studies. I note that Adora1 is highlighted and the Choi group as well as others (e.g. Erickson et al) have implicated astrocytes adenosine acting on Adora as important for alcohol actions and many other alcohol studies implicated astrocytes and microglia in transcriptome changes.

4. The authors used the CLUE - L1000 analysis of perturbagens (which is a nice addition to transcriptomics) but this is only briefly mentioned in the results and is not mentioned at all in the Discussion. Were the results not of interest or are there any problems with interpretation?

Reviewer #2: In the present manuscript, Radcliffe et al examine the genetic underpinnings of acute ethanol tolerance. They used 40 strains of the LXS reference population to explore gene by pre-treatment interactions. LXS RI strains were divided into two treatment groups receiving either saline or ethanol pretreatment 8 hours prior to sacrifice. The authors then used RNA-Seq to profile transcriptomic changes in whole brain (minus cerebellum) across the RI panel. Because they are working with RIs, they could then compare gene expression with LORR-related behaviors like sleep time and acute functional tolerance. They used a series of bioinformatics approaches to search for correlations between gene expression and other relevant phenotypes, measured differentially expressed genes, performed gene set enrichment analyses, and identified pre-treatment specific eQTLs. In addition, the authors identified a small number of genes that appear to be consistently expressed in response to ethanol across multiple, heterogeneous studies, and potential hub genes (Hdac1) that regulate ethanol response. Their results show both global and pre-treatment specific changes in gene expression. In doing so, they are able to begin to show the mechanism by with pre-treatment with EtOH influences sleep time and AFT. This will be of interest to the alcohol genetics community, especially those investigating level of response as an endophenotype for AUD.

Minor Weaknesses:

It is possible these results are sex specific, as only males were tested. But the benefit of working with RIs is that females can be tested at a later date and the data can be compared across studies.

Given that the authors observed widespread genetic effects on gene expression, this may be due in part to the fact that they were measuring gene expression in whole brain, rather than discrete brain regions or even cell-types.

Comments:

It was interesting that 72% of DE genes were also identified as DE in other studies by other lab groups. I think the genes listed in Table 1 will be valuable to the research community.

I may have missed this, but given that the number of significant DE genes varied across the RI strains, was this correlated at all with a given strain’s response to EtOH? Or was it only at the transcriptome rather than the behavioral level?

6. PLOS authors have the option to publish the peer review history of their article (what does this mean?). If published, this will include your full peer review and any attached files.

Reviewer #1: No

Reviewer #2: No

---

## [Author Response · Author response to Decision Letter 0]

14 Sep 2020

Dear Dr. Choi and Reviewers,

Thank you for taking the time to review our manuscript and providing us with an insightful critique. We present here for your consideration a revised manuscript that addresses the points you brought up as described in the following. Please note that the submission process for the raw and processed RNA-seq data is now complete and the data are now available in the Gene Expression Omnibus. This information along with the accession number is now included in the revision. 

Reviewer #1

This study provides new and potentially useful expression profiling of a large group of RI mice with and without alcohol treatment. The authors have done a great job of providing useful and detailed introduction and discussion for their work. 

We appreciate the kind comments. 

1. Some journals now require a section on limitations and caveats of the study and this one certainly needs such a discussion. Two major limitations are the use of whole brain and the small sample size (1, 2 or 3). All brain transcriptome studies show that brain region (and cell types) are major contributors of variance, which are lost here. And what are the implications of having n=1 for some samples?

We agree that discussion of the limitations of the study needs to be included and we have added a full paragraph that addresses this issue. Regarding the n=1 issue, please see the next comment.

2. Continuing with sample size, the Discussion states “The number of significant DE genes varied considerably by strain, from as few as 0 in several strains to over 500 in the most sensitive strain.” As I understand the methods, some of these comparisons are based on n=1, others on 2 or 3. How can a comparison with a single sample be statistically significant?

This is certainly a valid point, but there actually were no strains with an n of 1 in either pretreatment group as noted in the original Methods in the RNA extraction and library preparation section (second paragraph): “As a result, four strains (two in the saline group and two in the EtOH group) were left with an n of only one. These samples were also excluded leaving 40 strains in each of the pretreatment groups with n=2 and n=3 for 8 and 72 strain/pretreatment combinations, respectively.” In other words, all 80 groups had an n of either 2 or 3 and there was only a small number with n=2 in or one or both pretreatment groups (4 strains had n=2 in a single pretreatment group and 2 strains had n=2 in both pretreatment groups). 

Based on your concern, we took a closer look to determine the extent to which strains with n=2 influenced outcomes related to DE. Removal of these strains affected 68 of the 1,230 DE genes; approximately half of these were unannotated. We repeated the enrichment analyses using this set of 1,162 genes and the results were nearly identical to the original analyses. In addition, only 25 of the 887 genes found in common with other studies were lost and of the 15 genes listed in Table 1 that were significant in only a single LXS strain, only one was from a strain with n=2 (Cacna1g). 

Taken in isolation, an n of 2 is not ideal; however, we feel comfortable keeping these strains in the DE and related analyses given that they were conducted within the context of a large number of strains and because removing them had little effect on the downstream analyses. Also recall that each RNA-seq sample contained pooled RNA from three separate mice which helped to reduce random sources of variance. Finally, we want to remain consistent with the correlation and eQTL analyses which were dependent on strain mean and not strain variance. 

3. There is no mention of cell type except a bit of discussion about myelin. The data could be analyzed for some glial signatures, as has been done in other brain gene expression studies. I note that Adora1 is highlighted and the Choi group as well as others (e.g. Erickson et al) have implicated astrocytes adenosine acting on Adora as important for alcohol actions and many other alcohol studies implicated astrocytes and microglia in transcriptome changes.

Based on your suggestion, we investigated whether the DE genes or any of the four correlated gene sets significantly overlapped with expression signatures for neurons, astrocytes, or oligodendrocytes using the online tool Gene Set Enrichment Analysis (GSEA; www.gsea-msigdb.org/gsea/index.jsp). Unfortunately, the results were not especially revealing. The expression signatures for all three cell types significantly overlapped with the DE gene list while none of the signatures overlapped with any of the four correlated gene lists (FDR<0.05). We added these findings to the Results section and also addressed the issue in the new limitations paragraph in the Discussion. 

4. The authors used the CLUE - L1000 analysis of perturbagens (which is a nice addition to transcriptomics) but this is only briefly mentioned in the results and is not mentioned at all in the Discussion. Were the results not of interest or are there any problems with interpretation?

The original Discussion did mention the CLUE results, albeit only briefly and only in the context of findings related to Hdac1. We do believe that the CLUE findings may be of interest to investigators and are not necessarily uninterpretable, but we focused on the HDAC inhibitor results because of the very striking finding that the expression profiles of so many HDACi perturbagens (relative to the other PCLs in Table 5) were similar to those of only the DE genes and AFT-Et genes, and because Hdac1 was a high priority candidate gene based on previous studies from our lab. As interesting as the other PCLs are, we felt it would be cumbersome and perhaps a little forced to discuss all of them. We therefore focused on the HADCi findings which we believe was of most relevance to other findings in this study. Note that we have added an additional sentence in the Discussion related to the HDACi CLUE results, although, again, only in the context of Hdac1.

Reviewer #2 

It is possible these results are sex specific, as only males were tested. But the benefit of working with RIs is that females can be tested at a later date and the data can be compared across studies.

Indeed, it is possible if not likely that some of the results are sex specific. We address this in the new limitations paragraph (see critique 1 from Reviewer 1). We fully expect that future follow-up studies in this project will examine both sexes, whether in the LXS, other segregating populations, or in genetically modified lines. 

Given that the authors observed widespread genetic effects on gene expression, this may be due in part to the fact that they were measuring gene expression in whole brain, rather than discrete brain regions or even cell-types.

This may be true, although one could argue that genetic effects might actually be even more widespread in an analysis of discrete brain regions, or at least across multiple brain regions, because the most robust global genetic effects will have been observed in whole brain while a discrete structure analysis probably would have revealed those same robust effects as well as more subtle region-specific effects. This also is addressed in the new limitations paragraph. 

It was interesting that 72% of DE genes were also identified as DE in other studies by other lab groups. I think the genes listed in Table 1 will be valuable to the research community.

This finding was particularly satisfying given that there were substantial differences between our study and the published studies; i.e., species, dosing, brain regions, etc. It is certainly our hope that these results will be useful to other investigators.

I may have missed this, but given that the number of significant DE genes varied across the RI strains, was this correlated at all with a given strain’s response to EtOH? Or was it only at the transcriptome rather than the behavioral level?

This an interesting question and we did, in fact, examine those correlations, but chose not to include them in the original MS. Neither ST nor AFT in the saline pretreatment group correlated to the number of DE genes (unadjusted p>0.20) whereas both of the behaviors in the ethanol pretreatment group were significantly correlated to the number of DE genes (unadjusted p=0.01 for ST and p=0.04 for AFT). The fact that you brought this up suggests that this may be of interest to others and we have therefore added some information related to these findings in the Results and Discussion.

---

## [Editor Report · Decision Letter 1]

23 Sep 2020

Systems genetics analysis of the LXS recombinant inbred mouse strains: Genetic and molecular insights into acute ethanol tolerance

PONE-D-20-24132R1

Dear Dr. Radcliffe,

We’re pleased to inform you that your manuscript has been judged scientifically suitable for publication and will be formally accepted for publication once it meets all outstanding technical requirements.

Kind regards,

Doo-Sup Choi

Academic Editor

PLOS ONE
---

## [Editor Report · Acceptance letter]

5 Oct 2020

PONE-D-20-24132R1 

Systems genetics analysis of the LXS recombinant inbred mouse strains:Genetic and molecular insights into acute ethanol tolerance. 

Dear Dr. Radcliffe:

I'm pleased to inform you that your manuscript has been deemed suitable for publication in PLOS ONE. Congratulations! Your manuscript is now with our production department. 

Kind regards, 

on behalf of

Dr. Doo-Sup Choi 

Academic Editor

PLOS ONE